# Systematic Assessment of Transcriptomic Biomarkers for Immune Checkpoint Blockade Response in Cancer Immunotherapy

**DOI:** 10.3390/cancers13071639

**Published:** 2021-04-01

**Authors:** Shangqin Sun, Liwen Xu, Xinxin Zhang, Lin Pang, Zhilin Long, Chunyu Deng, Jiali Zhu, Shuting Zhou, Linyun Wan, Bo Pang, Yun Xiao

**Affiliations:** 1College of Bioinformatics Science and Technology, Harbin Medical University, Harbin 150081, China; sunshangqin@hrbmu.edu.cn (S.S.); xuliwen@hrbmu.edu.cn (L.X.); zhangxinxin@hrbmu.edu.cn (X.Z.); panglin@hrbmu.edu.cn (L.P.); zhujiali@hrbmu.edu.cn (J.Z.); shutingzhou58@gmail.com (S.Z.); wanlinyun@hrbmu.edu.cn (L.W.); 2School of Life Sciences, Westlake University, Hangzhou 310024, China; longzhilin@westlake.edu.cn; 3Wenzhou Research Institute, University of Chinese Academy of Science, Wenzhou 325001, China; dengchunyu54@gmail.com

**Keywords:** immune checkpoint blockade (ICB), transcriptomic biomarkers, immune response, immunotherapy, comparative analysis

## Abstract

**Simple Summary:**

The aim of our study was to evaluate the predictive performance of transcriptomic biomarkers to immune response. The study collected 22 transcriptomic biomarkers and constructed multiple benchmark datasets to evaluate their predictive performance of immune checkpoint blockade (ICB) response in pre-treatment patients with distinct ICB agents in diverse cancers. We found “Immune-checkpoint molecule” biomarkers PD-L1, PD-L2, CTLA-4 and IMPRES and the “Effector molecule” biomarker CYT showed significant associations with ICB response and clinical outcomes. These immune-checkpoint biomarkers and another immune effector IFN-gamma presented predictive ability in melanoma, urothelial cancer and clear cell renal-cell cancer. Interestingly, for anti-PD-1 therapy and anti-CTLA-4 therapy, the top-performing response biomarkers were usually mutually exclusive even though in the same biomarker category and most of biomarkers with outstanding predictive power were observed in patients with combined anti-PD-1 and anti-CTLA-4 therapy.

**Abstract:**

Background: Immune checkpoint blockade (ICB) therapy has yielded successful clinical responses in treatment of a minority of patients in certain cancer types. Substantial efforts were made to establish biomarkers for predicting responsiveness to ICB. However, the systematic assessment of these ICB response biomarkers remains insufficient. Methods: We collected 22 transcriptome-based biomarkers for ICB response and constructed multiple benchmark datasets to evaluate the associations with clinical response, predictive performance, and clinical efficacy of them in pre-treatment patients with distinct ICB agents in diverse cancers. Results: Overall, “Immune-checkpoint molecule” biomarkers PD-L1, PD-L2, CTLA-4 and IMPRES and the “Effector molecule” biomarker CYT showed significant associations with ICB response and clinical outcomes. These immune-checkpoint biomarkers and another immune effector IFN-gamma presented predictive ability in melanoma, urothelial cancer (UC) and clear cell renal-cell cancer (ccRCC). In non-small cell lung cancer (NSCLC), only PD-L2 and CTLA-4 showed preferable correlation with clinical response. Under different ICB therapies, the top-performing biomarkers were usually mutually exclusive in patients with anti-PD-1 and anti-CTLA-4 therapy, and most of biomarkers presented outstanding predictive power in patients with combined anti-PD-1 and anti-CTLA-4 therapy. Conclusions: Our results show these biomarkers had different performance in predicting ICB response across distinct ICB agents in diverse cancers.

## 1. Introduction

Cancer immunotherapies by immune checkpoint blockade (ICB) have revolutionized the conventional tumor treatment and bring remarkable clinical efficacy to patients of advanced-stage melanoma, squamous and non-squamous non-small-cell lung carcinoma (NSCLC), kidney carcinoma, DNA mismatch repair deficient (dMMR)/microsatellite instability (MSI) high cancers, gastric cancer and hepatocellular carcinoma [1,2]. However, the biggest limitation of ICB is that only a few patients are responsive to ICB due to intrinsic resistance to immunotherapy [3,4]. Therefore, predicting ICB response is a critical challenge for guiding patient selection for current checkpoint immunotherapies, and providing early on-treatment indicators of response.

Over the last decade, several dozens of biomarkers have been developed for predicting responsiveness to ICB [5,6], PD-L1 expression was one of the earliest and most promising biomarkers. Increasing studies found that pre-treatment PD-L1 expression on tumor cells and immune cells was associated with improved response rate, progression-free survival (PFS) and overall survival (OS), and could effectively predict the clinical outcomes to ICB across various tumor types [7,8]. Tumor mutation burden (TMB) emerged as a promising biomarker for ICB patient stratification. High mutation load correlated with increased response rate to ICB therapies and longer PFS in most of studies [9]. Among these biomarkers, transcriptome-based biomarkers that were generally composed of multiple genes were widely identified, such as immuno-predictive score (IMPRES), immune resistance program (IRP), pan-fibroblast TGF-β response signature (Pan-F-TBRS) [10], since the transcriptome sequencing of cancer patients was widespread and the transcriptomic changes were generally rapid following minimal damage. One of the most important advantages was that such multiple gene-based biomarkers could be effective to reflect immune-related properties with their expression levels. For instance, TIDE used the expression of multiple genes interacting with the cytotoxic T lymphocytes (CTL) infiltration level to reflect two immune escape mechanisms about the T cell dysfunction and exclusion. The results showed that TIDE achieved consistently better performance for both anti-PD-1 and anti-CTLA-4 therapies than other biomarkers like IFNG, PDL1 [11]. Immunophenoscore (IPS) that quantified tumor immunogenicity according to the expression of representative genes or gene sets was a useful biomarker of response to checkpoint blockades in patients with melanoma [12]. These predictive biomarkers underlay a diverse range of mechanisms of tumor-immune interaction implicating immune checkpoint pathway [13,14], immune regulation [15], tumor antigen presentation [16], lymphocyte infiltration [16,17] and tumor immune evasion [11]. However, the systematic assessment of predictive effects of these transcriptomic ICB response biomarkers is still absent.

Herein, we collected 22 transcriptomic biomarkers for ICB response, and determined a benchmark to evaluate the associations with immune response, predictive performance, and clinical efficacy of these transcriptomic biomarkers under different cancer types and ICB agents. Our results provided a systematic assessment of transcriptomic biomarkers for ICB response, which provided novel insights into potential strategies for patient stratification, and would help to improve the prediction accuracy in ICB immunotherapy.

## 2. Materials and Methods

### 2.1. Data Collection and Preprocessing

We collected gene expression profiles of 16 cancer types from the TCGA (https://gdc.cancer.gov/about-data/publications/pancanatlas, accessed on 6 October 2019), including 6518 tumor samples of bladder urothelial carcinoma (BLCA, *n* = 389), uterine corpus endometrial carcinoma (UCEC, *n* = 509), skin cutaneous melanoma (SKCM, *n* = 466), head and neck squamous cell carcinoma (HNSC, *n* = 476), prostate adenocarcinoma (PRAD, *n* = 444), cervical squamous cell carcinoma and endocervical adenocarcinoma (CESC, *n* = 300), colorectal (*n* = 344), lung squamous cell carcinoma (LUSC, *n* = 449), kidney renal clear cell carcinoma (KIRC, *n* = 460), liver hepatocellular carcinoma (LIHC, *n* = 319), breast invasive carcinoma (BRCA, *n* = 979), ovarian serous cystadenocarcinoma (OV, *n* = 301), glioblastoma multiforme (GBM, *n* = 153), lung adenocarcinoma (LUAD, *n* = 456), mesothelioma (MESO, *n* = 87), and stomach adenocarcinoma (STAD, *n* = 386).

For evaluation of ICB response prediction, we collected gene expression and the corresponding clinical data of ICB pre-treatment patients from ten studies (Table 1), which involved in four major cancer types (melanoma, ccRCC, UC, NSCLC) and four ICB therapy strategies (anti-CTLA-4, anti-PD-1, anti-PD-1 after progression on prior anti-CTLA-4 (anti-CTLA-4 prog anti-PD-1), and combination of anti-PD-1 and anti-CTLA-4 (combination therapy)). RNA-seq raw data of the Hugo et al. study [18], Riaz et al. study [19] and Gide et al. study [20] were available from GEO (GSE78220, GSE91061) and ENA (ERP105482), respectively. SRA archives were converted into fastq files with fastq-dump from SRA Toolkit v2.9.6, for the quality control of raw data, we used trim-galore (v.0.4.5) (https://github.com/FelixKrueger/TrimGalore, accessed on 18 October 2018) to filter low-quality reads. We used kallisto v0.44.0 (https://github.com/pachterlab/kallisto, accessed on 18 October 2018) [21] to align reads originating to the reference transcriptome and to quantify transcript abundance and then transformed count into gene-level. Expression levels were then converted into transcripts per million (TPM) and log2-transformed for downstream analyses. We collected the gene expression profiles from NanoString nCounter data for anti-PD-1 treatment samples from four cancer types (GSE93157). Other expression data and clinical information used in this study were obtained through the supplementary materials of original publications.

### 2.2. Biomarkers Collection and Scores Calculation

We performed an extensive search of published studies in the past five years about transcriptomic signatures for distinguishing patients who can benefit from ICI, on PubMed and Google Scholar. Keywords included “immune/immunotherapy”, “PD-1/PD-L1”, “CTLA-4”, “biomarker/signature”, “predict/predictor”, “checkpoint”, and “response”. Abstract and results sections (if necessary) were carefully scanned, biomarkers that met the following three criteria were included: (1) a biomarker is related to immune response or resistance; (2) a biomarker has predictive potential of immunotherapy response; (3) reaching more than 60 citations. The identified candidate biomarkers were then fully discussed by our group members. Finally, we collected a total of 22 transcriptomic biomarkers which were considered to be an ICB response indicator. For each biomarker, the score was calculated as follows: the expression of single checkpoint genes (PD-1, PD-L1, PD-L2 and CTLA-4), average gene expression-based biomarkers (CYT, gene.CD8, IFN-gamma, Expanded immune signature, EMT and CRMA) were applied in a straightforward manner. Another biomarker representing CD8 T cell infiltration was estimated via CIBERSORT (https://cibersort.stanford.edu/, accessed on 11 June 2019). IPS and IRP were computed using the source codes provided by the original publications. After data normalization, we used the TIDE web application for response prediction in each dataset (http://tide.dfci.harvard.edu/, accessed on 8 June 2019). Meanwhile, we reconstructed the calculation model for IMPRES and IS strictly according to the description in original publications, and proved that our calculation models were highly consistent with original articles using the same datasets (Appendix A). Other biomarkers were implemented using methods according to the original citations: TIS, APM, C-ECM-p were calculated using single-sample gene set enrichment (ssGSEA) analysis [27], and IPRES was calculated using gene set variation analysis (GSVA), both implemented in the GSVA R package [28]; Pan-F-TBRS was based on principal component analysis; T cell-inflamed GEP score was computed as the weighted sum of signature gene expressions. The abundance of eight immune and two stromal cell populations in the tumor microenvironment (TME) was estimated from bulk tissue gene expression data using the MCP-counter algorithm [29].

### 2.3. The Classification of Patients Based on Clinical Response

Four commonly used strategies to define responders and non-responders according to RECIST or irRECIST criteria were considered in our analysis: (1) PD (Progressive disease) strategy: patients with complete or partial response, or stable disease were defined as “responders”. Patients with progressive disease were defined as “non-responders” [18,30]. (2) OR (Objective response) strategy: the “responders” were defined using a composite end point of complete response or partial response by RECIST criteria. “non-responders” were defined as stable or progressive disease [15]. (3) OS (No long overall survival response) strategy: “responders” were defined using a composite end point of complete or partial response by RECIST criteria or stable disease with overall survival greater than one year. “non-responders” were defined as progressive disease by RECIST criteria or stable disease with overall survival less than one year [13]. (4) DCB (Durable clinical benefit) strategy: “responders” were defined using a composite end point of complete or partial response by RECIST criteria or stable disease with progression free survival greater than six months. “non-responders” were defined as progressive disease by RECIST criteria or stable disease with progression free survival less than six months [31].

### 2.4. Evaluation of the Predictive Performance of Biomarkers

The AUCs were calculated to evaluate the predictive abilities of biomarkers to clinical response based on “PD” strategy. Receiver operating characteristic (ROC) curves were constructed using the function *roc* provided in the pROC package [32]. For each biomarker, its training data was not included in the AUC calculation. Considered different sample size of datasets, we computed the aggregated sample size-weighted AUC (labeled as Prediction Score) for comparing the different biomarkers regardless the data size [33,34]. The Prediction Score for one biomarker is defined as follows:$$PredictionScore=\frac{\sum_{i=1}^{n}Size_{i}\times AUC_{i}}{\sum_{i=1}^{n}Size_{i}}$$
where *AUC_i_* is the prediction performance of this biomarker in *i*-th dataset, and *Size_i_* is the sample size of the *i*-th dataset.

### 2.5. Survival Analysis

Patients who had more than one sample were excluded in survival analysis. Kaplan–Meier analyses were performed to estimate the association of biomarker scores with overall survival (OS) and progression free-survival (PFS) between patients with high scores (>median) and those with low scores (<median) (log-rank tests) using the survival package. The patients with median score (=median) were classified into the smaller-size group among the two groups mentioned above. *p*-value < 0.05 was considered as a significant difference.

### 2.6. Objective Response Rate

We assessed the objective response rate in the objective response evaluable datasets, containing more than 20 intention-to-treat patients who had measurable disease according to RECIST at baseline. Such that patients with a complete or partial response were defined as objective response patients. Objective response rate (ORR) was defined as the number of objective response patients divided by the total number of patients. Fisher’s exact test was used to assess the association between biomarker scores and objective response.

### 2.7. Statistical Analysis

We tested the ability of the 22 transcriptomic biomarkers to predict clinical response by comparing (two-sided Wilcoxon Rank Sum Test) between the responders and non-responders. When the *p* value was less than 10^−5^, we uniformly set it to *p* = 10^−5^. Logistic regression modeling was used to conduct the hypothesis testing associated with these biomarkers and clinical response outcome under ICB therapies.

## 3. Results

### 3.1. Collection of Immune Response-Associated Transcriptomic Biomarkers

We curated 22 transcriptomic biomarkers for ICB response that were based on the expression of a single gene or multiple genes involving in the immune-associated processes such as lymphocyte infiltration, antigen presenting, immune resistance and immune escape (Figure 1 and Table 2). We classified these biomarkers into seven categories: (i) “Immune-checkpoint molecule”: PD-1, PD-L1, PD-L2, CTLA-4 and IMPRES; (ii) “Tumor-infiltrating lymphocyte (TIL)” category representing the tumor infiltration level of CD8+ T cell and other immune cell types: CIBERSORT.CD8, gene.CD8 and TIS; (iii) “Effector molecule” category associated with cytolytic activity and IFN-γ responsive effect: CYT, IFN-gamma, Expanded immune signature, and T cell-inflamed GEP; (iv) “Antigen associated” category: CRMA as the gene expression signature related to MAGE-A cancer-germline antigens that predicted resistance uniquely to CTLA-4 blockade; (v) “Antigen presenting” category: Antigen presenting machinery (APM); (vi) “Immune resistance” category shaping the tumor microenvironment to restrain anti-tumor immunity: the Pan-F-TBRS, EMT, C-ECM-up, IRP and IPRES; (vii) “Comprehensive” category involving multiplexed process of anti-tumor immunity: TIDE, IPS and IS.

Some of these biomarkers contained single gene (e.g., PD-1), others contained multiple immune-related genes (e.g., IMPRES and TIS). Although a few common genes were shared among these biomarkers, 67.4% of the biomarkers did not share any overlapping genes (Appendix A). The majority of multiple-gene biomarkers did not contain known immune checkpoint genes, such as PD-1, PD-L1, PD-L2, and CTLA-4. However, significant overlaps among IS, IPRES, TIS and IPS were observed (*p* < 0.01, Hypergeometric test), having a common gene *LCK* among them. To further investigate the correlations between these biomarkers, we calculated the scores of the transcriptomic biomarkers for ~6600 cancer samples derived from the TCGA pan-cancer cohort and calculated their similarities using Spearman rank correlation (See Method). There were two distinct clusters. The biomarkers from “Immune-checkpoint molecule”, “Tumor-infiltrating lymphocyte (TIL)”, “Effector molecule” and “Antigen presentation” categories formed one cluster with average correlation coefficient of 0.711 (Figure 2). The biomarkers including Pan-F-TBRS, IPRES and C-ECM-up of “Immune resistance” category formed another cluster, showing an average Spearman’s rank-correlation coefficient of 0.757 (Figure 2). The other three biomarkers, IPS, CIBERSORT.CD8 and Mutation_load, were not contained in these two clusters. Notably, all of the transcriptomic biomarkers showed low correlations with Mutation_load, implying different anti-tumor immune response mechanisms from TMB.

### 3.2. Assembling Benchmark Datasets to Test Transcriptomic Biomarkers

To perform an overall evaluation of the correlation between the predictive biomarkers and clinical response to ICB therapy, we compiled numerous public ICB treatment benchmark datasets derived from 10 studies containing a total of 647 patients with solid tumors involving in melanoma, clear cell renal-cell cancer (ccRCC), urothelial cancer (UC) and non-small cell lung cancer (NSCLC) under different ICB therapies (Table 1). Among these patients, the overall objective response rate was 28.4% across different cancer types. For melanoma, ccRCC, UC and NSCLC, their objective response rates were 35.2%, 24.2%, 23.6% and 25.7%, respectively. The melanoma patients showed a higher objective response rate to anti-PD-1 therapy (42.2%) and the anti-PD-1 and anti-CTLA-4 combination therapy (65.6%) (Appendix A). The benchmark datasets we used supplied additional validation data to the reference studies (Appendix A).

In order to facilitate subsequent evaluation, we applied a widely used classification strategy for defining the responding and non-responding sub-groups according to the RECIST criteria or irRECIST criteria. Patients who achieved objective responses (complete response or partial response), or stable disease were classified as responders, and non-responders were defined as progressive disease. The total number of responders and non-responders were 316 and 309, respectively. In addition, the other response classification strategies, including “OR” (Objective response), “OS” (No long overall survival response), and “DCB” (Durable clinical benefit) that were frequently used in previous studies, were also applied for classifying patients to investigate whether different classification strategies could affect the evaluation results.

### 3.3. Assessing the Association between Transcriptomic Biomarkers and Clinical Response in Benchmark Datasets

Based on these benchmark datasets, we addressed whether the predictive values of transcriptomic biomarkers could differentiate responders from non-responders to ICB by wilcoxon rank sum test. And we also performed univariate logistic regression analysis to test associations between response and these biomarkers. In general, these biomarkers had remarkably different associations with clinical response in different patient cohorts. Among them, the immune checkpoint molecules PD-1, PD-L1, PD-L2 and CTLA-4 showed the most stable correlations, as their expression levels were significantly higher in responding sub-groups across three individual cohorts (Appendix A, *p* < 0.05 for all, Wilcoxon Rank Sum Test). CYT was another stable biomarker whose score was significantly higher among the responders in three cohorts (Appendix A, *p* = 0.031 in the Prat et al., 2017, *p* = 0.004 in the Gide et al., 2019, *p* = 0.004 in the Mariathasan et al., 2018, Wilcoxon Rank Sum Test). Its positive correlations with response were also true when assessed by logistic regression analysis (Appendix A, *p* = 0.011, odds ratio = 2.08, [95% CI, 1.24–3.86] in the Gide et al., 2019; *p* = 0.031, odds ratio = 1.62, [95% CI, 1.07–2.58] in the Prat et al., 2017). High scores of IFN-gamma, Expanded immune signature, T cell−inflamed GEP, genes.CD8, TIS, APM and IS were all significantly associated with clinical response in the two datasets of the Gide et al., 2019 and the Mariathasan et al., 2018 studies (Appendix A, *p* < 0.05 for all, Wilcoxon Rank Sum Test). Unlike single checkpoint molecules, IMPRES encompassing several pairs of checkpoints was positively correlated with response in another two cohorts, the Van Allen et al., 2015 and the Chen et al., 2016 ones (Appendix A, *p* = 0.013, *p* = 0.037, respectively, Wilcoxon Rank Sum Test; *p* = 0.019, odds ratio = 2.14, [95% CI, 1.21–4.46], *p* = 0.033, odds ratio = 2.05, [95% CI, 1.15–4.51], respectively, Logistic regression model).

Interestingly, we found several biomarkers presented significant correlations with clinical response in one specific patient cohort. Biomarkers indicating immune resistance like EMT, C−ECM−up and Pan−F−TBRS had significantly lower scores in responders only from the Hugo et al., 2016 dataset (Appendix A, *p* < 0.05 for all, Wilcoxon Rank Sum Test), their negative correlations with response were also confirmed by logistic regression analysis (Appendix A, *p* = 0.01, odds ratio = 0.02, [95% CI, 3 × 10^−4^ − 0.22], *p* = 0.013, odds ratio = 0.01, [95% CI, 10^−4^ − 0.22], *p* = 0.03, odds ratio = 0.64, [95% CI, 0.4–0.91] in the Hugo et al., 2016 dataset, respectively). Previous study showed most of the non-responders in the Hugo et al., 2016 cohort displayed innate anti-PD-1 resistant characteristics which might lead to the primary associations between the “Immune resistance” biomarkers and ICB response [18].

We further observed differences of biomarkers between responders and non-responders under the other three classification strategies (i.e., “OR”, “OS” and “DCB”). In general, the trends of associations between biomarkers and clinical response didn’t change as strategies changed. We also observed that even under all of the four response classification strategies, there were still some datasets that did not capture any correlation between biomarkers and clinical response (Appendix A), suggesting current biomarkers might still have some limitations and could not make good use for all sample sets.

### 3.4. The Prediction Performance of Transcriptomic Biomarkers for Response to ICB Immunotherapy

To evaluate the prediction performance of the transcriptomic biomarkers for ICB response, receiver operator characteristic (ROC) curves were performed to measure the true-positive rates against the false positive rates in benchmark datasets. According to the average AUC across datasets, the widely used ICB response biomarkers TMB and PD-L1 were superior to most of biomarkers, though they didn’t achieve good performance (Appendix A, average AUC 0.66 for Mutation_load, average AUC 0.65 for NonSyn_mutation_load, average AUC 0.63 for PD-L1). IMPRES ranked among the top two with the AUC being above 0.7 in four datasets. Appendix A showed the ROC curves of PD-L1 and IMPRES across all benchmark datasets. APM, Pan-F-TBRS and CYT ranked in the top fourth, sixth and seventh place (Appendix A, average AUC 0.63, 0.62, 0.62 for APM, Pan-F-TBRS and CYT, respectively). TIDE and C-ECM-up followed closely, while the performance fluctuated noticeably among different datasets (Appendix A, average AUC 0.61 for TIDE, average AUC 0.61 for C-ECM-up). Also, some of the biomarkers had lower average AUC, but in some specific datasets their prediction performances were extremely high, such as EMT in the Hugo et al., 2016 (Appendix A, AUC = 0.82). Using the sum of sample size-weighted AUC (labeled as Prediction Score), we summarized the prediction performance of transcriptomic biomarkers acorss datasets (Appendix A), the training sets used for identification of biomarkers were excluded to ensure an unbiased results. All of the biomarkers had the prediction score lower than 0.7.

In terms of sensitivity and specificity, we used median score of biomarkers as the threshold to stratify responders and non-responders. Among all the biomarkers, the median sensitivities and median specificities of IS, Mutation_load and CRMA were consistently high. IMPRES showed the highest median sensitivity to response but very low median specificity. In contrast, IPS and TIDE had the lowest sensitivities, but their specificities were relatively higher (Appendix A). In summary, the predictive ability of these transcriptomic biomarkers to ICB response is still limited currently.

### 3.5. Evaluating the Association of Biomarkers with Clinical Responses in Specific Cancer Types with Different ICB Therapies

#### 3.5.1. Evaluating the Association in Different Cancer Types

Due to highly heterogeneous objective response rate in different cancer types (melanoma, 31–44% [44,45,46]; NSCLC, 19–20% [47,48]; RCC, 22–25% [49,50]) (Appendix A), we therefore asked whether correlations between these biomarkers and response to ICB were different in various cancer patients under the setting of different ICB therapies. We first splited the 10 benchmark datasets into 15 patient cohorts according to different cancer types, including melanoma (10 datasets, *n* = 268), ccRCC (2 datasets, *n* = 33), UC (2 datasets, *n* = 324) and NSCLC (1 dataset, *n* = 35), and then performed the response association analysis as above (Table 1).

Different biomarkers showed differential correlations with response in various cancer types, and some had the preference in specific cancer types. Across cancer types, only the biomarkers of “Immune-checkpoint molecule” category performed stably in four cancer types. CYT, IFN-gamma, gene.CD8, APM and IS presented significantly higher scores among responders of three cancer types except for NSCLC (Figure 3 and Appendix A, *p* < 0.05 for all, Wilcoxon Rank Sum Test). Melanoma and UC shared the largest number of biomarkers correlated with response, including PD-1, PD-L1, PD-L2, CTLA-4, gene.CD8, all biomarkers of “Effector molecule”, APM and IS (Figure 3A,C). On the other hand, transcriptomic biomarkers presented preference in specific cancer types. Among biomarkers of “Immune-checkpoint molecule”, PD-L1 showed significant associations in ccRCC (Figure 3B, *p* = 0.004 in the Miao et al., 2018 data (ICB)) while PD−L2 and CTLA−4 was significant in NSCLC (Figure 3D, Appendix A, *p* = 0.024, *p* = 0.028 in the Prat et al., 2017(NSCLC) dataset, respectively, Wilcoxon Rank Sum Test; *p* = 0.02, odds ratio = 2.9, [95%CI, 1.31–8.33], *p* = 0.04, odds ratio = 1.94, [95%CI, 1.07–3.88], respectively, Logistic regression model). Specifically, CIBERSORT.CD8 only showed significant associations with clinical response in ccRCC (Figure 3B, *p* = 0.027 in the Miao et al., 2018 study (ICB), Wilcoxon Rank Sum Test) and IPS only in UC patients (Figure 3C and Appendix A, *p* = 0.006 in the Mariathasan et al., 2018 report, Wilcoxon Rank Sum Test). The immune resistance-related biomarkers Pan-F-TBRS, EMT and C-ECM-up exhibited a significant correlation with poor clinical response only in the melanoma patients (Figure 3A and Appendix A, *p* < 0.05 for all, Wilcoxon Rank Sum Test). Based on the other three response classification strategies, the overall trends of the results didn’t change remarkably (Appendix A).

#### 3.5.2. Evaluating the Association in Different ICB Therapy Strategies

To further explore the appropriate biomarkers for different ICB treatments, we evaluated the correlations between these transcriptomic biomarkers and response in ten melanoma cohorts under different ICB therapy strategies including anti-PD-1, anti-CTLA-4, anti-PD-1 after progression on prior anti-CTLA-4 (anti-CTLA-4 prog anti-PD-1) and combined anti-PD-1 and anti-CTLA-4 immunotherapy (combination therapy) (Table 1).

For the patients treated with anti-PD-1, checkpoint PD-L1, gene.CD8, immune resistance-related biomarkers (Pan-F-TBRS, EMT and C-ECM-up) and TIDE demonstrated significant correlations with clinical response in the Gide et al., 2019 (aPD1) or the Hugo et al., 2016 datasets (Figure 4A). The consistent results were observed in logistic regression analysis (Appendix A). In the anti-CTLA-4 cohorts, PD-L2, CTLA-4 and IMPRES biomarkers in “Immune-checkpoint molecule” category showed correlations with ICB response only in one of three datasets, which might be caused by the limited sample size of the other two datasets (Figure 4B and Appendix A, *p* = 0.049, *p* = 0.049, *p* = 0.013 in Van Allen et al., 2015, respectively, Wilcoxon Rank Sum Test). For the “anti-CTLA-4 prog anti-PD-1” therapy, PD-L2, IMPRES, effector molecules (CYT, Expanded immune signature) and TIDE all had significant correlations with response in the Chen et al., 2016 data (aCTLA4 prog aPD1) (Figure 4C and Appendix A, *p* < 0.05 for all, Wilcoxon Rank Sum Test). Comparing with other ICB therapies, the combination therapy dataset captured the largest number of biomarkers that significantly associated with improved clinical response, and these biomarkers included the checkpoint molecules (PD-1, PD-L1, CTLA-4), T-cell infiltration-related signature (TIS), effector molecules (CYT, IFN-gamma, Expanded immune signature), APM and IS (Figure 4D and Appendix A).

Across these ICB therapies, the biomarkers from “Immune-checkpoint molecule” had the most extensive applicability, whose scores were significantly higher in the responders of four different immunotherapy datasets. The biomarkers belonging to “Comprehensive” represented significant associations with ICB response in three ICB therapy strategies (including anti-PD-1, anti-CTLA-4 prog anti-PD-1 and combination therapy). Among the biomarkers of “Immune-checkpoint molecule”, PD-L1 showed significant association with response in patients treated with anti-PD-1 or combination therapy, while PD-L2 and IMPRES showed significant associations in patients treated with anti-CTLA-4 or “anti-CTLA-4 prog anti-PD-1” therapy. Scores of APM biomarker were significantly higher in the responders treated with anti-PD-1 and combination therapy. The overall trend of the results did not change significantly when we used the other three response classification strategies (Appendix A).

### 3.6. Evaluating Prediction Performance of Transcriptomic Biomarkers in Specific Cancer Types and ICB Therapies

We next evaluated the prediction performance of the transcriptomic biomarkers in the 15 cancer type-specific datasets (Table 1). In melanoma, the performance of Mutation_load, IFN-gamma, Nonsyn_mutation_load, PD-L1 and Expanded immune signature were superior to the other biomarkers (Figure 5A, Appendix A, mean AUC = 0.68, 0.67, 0.66, 0.65, 0.65, respectively). For ccRCC patients, we found that the biomarkers CIBERSORT.CD8, gene.CD8 and the T cell-inflamed GEP, IFN-gamma, CYT and PD-L1 provided the relatively better performance in the two cohorts (Figure 5B and Appendix A, mean AUC = 0.8, 0.78, 0.78, 0.76, 0.75, 0.73, respectively). Among the UC patients, beside Neoantigen_load that came out at top, the transcriptomic biomarkers Pan-F-TBRS and APM ranked among the top two and top three though the average AUC merely reached 0.65 (Figure 5C and Appendix A). While for NSCLC, PD-L2 and CTLA-4 obtained relatively higher accuracy in Prat et al., 2017 (Figure 5D and Appendix A, AUC = 0.73 for PD-L2 and AUC = 0.72 for CTLA-4). In general, the overall predictive efficacy of transcriptomic biomarkers was higher in ccRCC across four cancer types. Some biomarkers preferred the specific cancer in predicting response to ICB, for example, PD−L1 and IFN−gamma performed well in melanoma in contrast to their worse performance in NSCLC (Figure 5A,D).

Furthermore, we investigated the response predictive performance of these biomarkers under different ICB therapy strategies in 10 melanoma datasets (Table 1). Firstly, we focused on patients with anti-PD-1 or anti-CTLA-4 monotherapy. Only the NonSyn_mutation_load was shared as the biomarker with high predictive efficiency in both of these two therapies. The top five transcriptomic biomarkers ranked according to the average AUC in the anti-PD-1 treatment dataset were C-ECM-up, TIDE, CYT, PD-L1 and PD-1, however, these five biomarkers couldn’t achieve effective accuracy in anti-CTLA-4 therapy with average AUC around 0.5 (Figure 5E). Alternatively, IMPRES, IFN-gamma and IPS obtained top-ranked average AUC in anti-CTLA-4 therapy, but their predictive performance in anti-PD-1 therapy was poor (Figure 5E,F). One possible explanation was that the PD-1 and CTLA-4 pathways drove distinct immunobiologic processes [51,52]. Next, we found that in “anti-CTLA-4 prog anti-PD-1 therapy” cohorts, Expanded immune signature, TIDE and IFN-gamma achieved relatively better performance with the AUC greater than 0.75 (Figure 5G, average AUC = 0.79, 0.78, 0.76, respectively). And in patients treated with combined anti-CTLA-4 and anti-PD-1 therapy, the biomarkers PD-L1, TIS, APM and IFN-gamma ranked in the top four, all with the AUC being above 0.8 (Figure 5H, average AUC = 0.83, 0.82, 0.81 and 0.81, respectively). According to the prediction score, we may have a glance at the overview of preference for biomarkers. PD−L1 and CYT were more predictive in combined anti-CTLA-4 and anti-PD-1 therapy and anti-PD-1 monotherapy (Appendix A).

### 3.7. The Effect of Response Biomarkers on Clinical Efficacy of ICB Therapy

To investigate the impact of response biomarkers’ status on predicting clinical outcome to ICB therapy, the Kaplan-Meier analyses of overall survival (OS) and progression-free survival (PFS) were performed. At the overall level, the response biomarkers of “Immune-checkpoint molecule” category displayed predictive natures for ICB immunotherapy, which were superior to the other biomarkers (Appendix A). Within the full cohort from the Van Allen et al. study, patients with high CTLA-4 expression had improved overall and progression-free survival than those with low CTLA-4 expression (Appendix A, log-rank *p* = 0.0065 and *p* = 0.027, respectively). Median OS was 38.7 weeks (95% CI, 16.6–115.9) and the 24-week landmark overall survival rate was 52.4% (95% CI, 34.8–78.8) for CTLA-4–low patients compared with 169.1 weeks (95% CI, 30.1 to not estimable) and 76.2% (95% CI, 60–96.8) for CTLA-4–high patients (Appendix A). Similarly, a statistically significant association of detectable PD-L2 status with OS and PFS was observed in both the Gide et al., 2019 (Appendix A, log-rank *p* = 0.03 and *p* = 0.017) and Miao et al., 2018 datasets (Appendix A, log-rank *p* = 0.0087 and *p* = 0.037). Meanwhile, the biomarkers Expanded immune signature, T cell-inflamed GEP of “Effector molecule” category and the IPS of “Comprehensive” category suggested preferable prognostic ability under the setting of ICB therapy (Appendix A).

We next explored the predictive ability of response biomarkers on clinical outcomes to ICB immunotherapy across different cancer types. Among melanoma tumors, the biomarkers PD-L1, PD-L2 and CTLA-4 and those of “Effector molecule” category, such as Expanded immune signature and T cell-inflamed GEP, were significantly associated with favorable outcomes (Figure 6A, Appendix A). In contrast, the values of CRMA were correlated with poor overall and progression-free survival observed in the Van Allen et al., 2015 dataset (Appendix Ab, log-rank *p* = 0.0013 and *p* = 0.0058). In UC, the results showed that PD-L1, CTLA-4, IFN-gamma, Expanded immune signature and T cell-inflamed GEP in “Effector molecule” category, IPS and IS were positively correlated with longer overall survival (Figure 6A, Appendix A, log-rank *p* < 0.05 for all), whereas higher EMT signature scores alone was associated with worse progression-free survival (Appendix A, log-rank *p* = 0.045). For ccRCC, the only statistically significant association of PD-L2 values with OS was observed in the Miao et al., 2018 (aPD1) dataset (Figure 6A, Appendix A, log-rank *p* = 0.012). However, we did not find any biomarkers being significantly associated with survival outcome in NSCLC patients (Appendix A). These results suggested that different biomarkers displayed different predictive ability to ICB response in specific cancer type. Of note, the “Immune-checkpoint molecule” biomarkers showed relatively stable clinical efficacy to ICB therapy across multiple cancer types.

Furthermore, we evaluated the prognostic significance of response biomarkers under different ICB therapy strategies in melanoma. For anti-PD-1 therapy, patients with high scores of PD-L1, gene.CD8, APM, IS and all biomarkers in “Effector molecule” category showed significantly favorable clinical outcome, which were mainly observed in the Gide et al., 2019 (aPD1) dataset (Figure 6B, Appendix A). For anti-CTLA-4 blockade, we observed that higher CTLA-4 expression and higher scores of TIS and Expanded immune signature were associated with better OS and PFS, whereas the values of CRMA and TIDE were associated with worse survival (Figure 6B, Appendix A). These complementary findings between PD-1 and CTLA-4 therapy were accordant with the notion that the PD-1 and CTLA-4 pathways occupied biologically and clinically disparate niches. Considering patients treated with CTLA-4 blockade followed by PD1 antibodies, the positive association between IPS and overall survival was found (Figure 6B, Appendix A, log-rank *p* = 0.036). For the combination anti-PD-1 and anti-CTLA-4 immunotherapy, only the higher value of APM observed was significantly associated with increased progression-free survival (Figure 6B, Appendix A, log-rank *p* = 0.037).

Next, we asked whether the different scores of ICB response biomarkers resulted in differential objective response to ICB therapy. Objective response rate (ORR) was evaluated based on the median of biomarker scores as a differentiating threshold in multiple datasets at the different situation, in which the number of samples was more than 20 to ensure reliability. At the overall level, the scores of PD-L2, gene.CD8 and the biomarkers in “Effector molecule” category were positively correlated with objective response to ICB (Figure 6C and Appendix A). Of these, the ORRs in the PD-L2–high expression patients (80.6%) were numerically greater than those in the PD-L2–low expression patients (35.1%) in the Gide et al., 2019 dataset (Figure 6C). Nevertheless, higher scores of Pan-F-TBRS and IPRES in “Immune resistance” category were associated with lower objective response rates (Figure 6C and Appendix A). Furthermore, the objective response in patients with different biomarker scores was assessed, considering different cancer types (involving melanoma, UC and NSCLC) and different ICB therapy strategies. For melanoma, objective response rates were higher in patients with high score of PD-L2, TIS, IFN-gamma, T cell-inflamed GEP and APM (Appendix A). For UC, the scores of Expanded immune signature, IPS and Pan−F−TBRS were significantly associated with clinical response to ICB, whereas none of the predictive biomarkers in NSCLC was significantly correlated with objective response rate (Appendix A). Among different ICB therapy strategies, we observed almost all biomarkers were associated with objective response to PD-1 blockade in at least one dataset (Appendix A). Meanwhile, the impact of the statuses of PD-L2, TIS, IFN-gamma, T cell-inflamed GEP, AMP and IRP on objective response was also observed in melanoma patients with combination anti-PD-1 and anti-CTLA-4 immunotherapy (Appendix A). And there were no significant correlations observed between the scores of any predictive biomarker and clinical response to other immunotherapy strategies in melanoma patients (Appendix A). Taken together, these data suggested response biomarkers as transcriptomic determinants of clinical outcome depended on different situations relating to cancer types and immunotherapy strategies.

### 3.8. Evaluating the Association of Tumor Microenvironment Components with Response to ICB Immunotherapy and Their Prediction Performance

As various TME components can influence response and resistance to ICB immunotherapy [53], we evaluated whether the abundance of TME components estimated by MCP-counter associated with ICB clinical response [29].

Different TME components also showed distinct correlations with ICB response in different cancer cohorts following various ICB therapy strategies. Among these, CD8 T cell and T cell showed the most significantly positive correlations with response to ICB immunotherapy across melanoma, ccRCC and UC. Notably, high abundances of CD8 T cells, T cells and myeloid dendritic cells were significantly associated with clinical response in melanoma treated with combined anti-CTLA-4 and anti-PD-1 therapy, while in ccRCC, these cell populations showed positive correlation with response for patients treated with anti-PD-1. For UC patients, the abundances of CD8 T cells, T cells, cytotoxic lymphocytes and especially B lineages were significantly higher in the responders treated with anti-PD-L1 therapy. This was consistent with recent reports that B cells were associated with immunotherapy response to ICB in patients with metastatic melanoma, RCC and soft-tissue sarcoma [54,55]. Moreover, we also found that the stromal composition including fibroblasts and endothelial cells had significantly lower abundance score in responders only from the Hugo et al., 2016 dataset (Figure 7A, *p* < 0.05 for all, Wilcoxon Rank Sum Test).

Furthermore, the prediction performance of these TME components was evaluated correspondingly. In melanoma, the CD8 T cell and NK cell had the relatively better performance (prediction score = 0.64, 0.62, respectively). Among ccRCC patients, we found that CD8 T cell, T cell, cytotoxic lymphocytes, myeloid dendritic cells and B lineage all achieved relatively better performance with the AUC greater than 0.70 (prediction score = 0.80, 0.71, 0.76, 0.70, 0.71, respectively). For UC, only the cytotoxic lymphocytes ranked in the top with the prediction score being 0.62. Next, we explored the performance of TME components in melanoma under different ICB therapy strategies. NK cell was relatively more effective for predicting response to anti-PD-1 therapy, while B lineage obtained relatively better performance for melanoma patients under anti-CTLA-4 therapy. In patients treated with combined anti-CTLA-4 and anti-PD-1 therapy, the majority of cell populations achieved high prediction score especially myeloid dendritic cells, CD8 T cell and T cell (all greater than 0.75) (Figure 7B).

## 4. Discussion

Immune checkpoint blockade therapy has prompted a shift of therapeutic landscape and induce durable responses in several advanced-stage cancers, but only a fraction of patients benefits from the treatment. Identifying ICB response biomarkers is a crucial mandate for successful clinical application of these agents [1,5,6,56]. In this study, we systematically evaluated predictive power of current 22 transcriptomic biomarkers for ICB response, which involved in immune checkpoint, lymphocyte infiltration, immune resistance, immune escape and other mechanisms, using multiple ICB treatment benchmark datasets constructed for different evaluation situations. The scores of each response biomarker were assessed carefully and their calculation process could be reproduced well following the original publications in our study.

Our results suggested that, for the overall evaluation, the biomarkers PD-1, PD-L1, PD-L2, CTLA-4 and IMPRES of “Immune-checkpoint molecule” and CYT of “Effector molecule” showed the most stable correlations with ICB response and outstanding prediction performance. CYT quantified the cytolytic activity of the local immune infiltrate based on transcript levels of two key cytolytic effectors, granzyme A (GZMA) and perforin (PRF1), which represented the process of killing cancer cells that tumor immunotherapy strategies aim to boost [57]. Accumulating evidence supported CYT as an impactful prognostic feature of tumors and was considered widely as an immunotherapy response indicator. CYT was associated with a modest but significant survival benefit in pan-cancer samples and commonly in melanoma [39,58,59]. Also, patients who achieved clinical benefit from ICB therapy had significantly higher CYT than those who had minimal benefit from ICB therapy [22]. We also found that PD-L1, PD-L2, CTLA-4 and the biomarkers Expanded immune signature, T cell-inflamed GEP in “Effector molecule” category could also be a transcriptomic determinant of clinical outcome and were associated with the objective clinical response to ICB. To evaluate the performance of response biomarkers, we also calculated the accuracy, sensitivity and specificity of 22 biomarkers. TMB-associated biomarkers were included in the evaluation comparison, which were often reported to indicate the likelihood of response to ICB immunotherapy [60,61]. As expected, the performance of Mutation_load and NonSyn_mutation_load in TMB category ranked among the top ones. Compared with TMB biomarkers, IMPRES of “Immune-checkpoint molecule” and CYT of “Effector molecule” both demonstrated relatively good predictive performance and high sensitivity, though their specificities were actually barely satisfactory. These findings suggested that the “Immune-checkpoint molecule” and the “Effector molecule” biomarkers might play a role on predictive responses and clinical outcomes with ICB therapy, consistent with many previous studies, which demonstrated their robustness and generalizability [8,14,59].

Interestingly, we observed that in some benchmark datasets, most of biomarkers showed robust performance for ICB response prediction, whereas none in other datasets. Meanwhile, the biomarkers Pan−F−TBRS, EMT, and C−ECM−up of “Immune resistance” category showed differential expression between responders and non-responders only in the Hugo et al., 2016 dataset. Indeed, previous study reported that non-responders in the Hugo et al., 2016 displayed innate anti-PD-1 resistant characteristics represented by the transcriptional signature IPRES, which indicated concurrent up-regulation of processes involving in the mesenchymal transition, extracellular matrix remodeling, and angiogenesis [18]. On the other hand, TGF-β-signaling, EMT and remodeling of the extracellular matrix (ECM) laid down by fibroblasts often occurred in tumors with T cell suppression or exclusion [10,30,41]. It seemed to be the innate characteristics of patients in Hugo study, which led to the primary associations between the “Immune resistance” biomarkers and ICB response. These results promoted us to ask whether the performance of biomarkers to predict ICB response depended on the inherent nature of samples, such as the state of immune infiltration and the cancer-immune phenotypes [62]. And it also reminded us to evaluate the response biomarkers under different situations including different cancer types and ICB therapy strategies.

Next, we explored the robustness of biomarker to predict ICB response across different cancer types including melanoma, UC, ccRCC and NSCLC. In melanoma, most biomarkers were significantly associated with clinical response. Indeed, most of the transcriptomic biomarkers for ICB response have been designed, identified or validated in melanoma [63,64]. Specially, we found the checkpoint molecules PD-L1, PD-L2, CTLA-4, the immune effector IFN-gamma, Expanded immune signature, T cell-inflamed GEP and the biomarker APM showed significantly higher scores in responders with ICB blockade, and they were also consistently correlated with clinical outcome and objective clinical response to ICB. For UC, the results suggested that PD-L1, CTLA-4, the biomarkers IFN-gamma, Expanded immune signature, T cell-inflamed GEP of immune effector molecules, and the biomarkers IPS, IS in “Comprehensive” category showed better impact on response association and clinical efficiency than other biomarkers. Previous studies reported that IFN-gamma and Expanded immune signature performed better predictive ability than most of other biomarkers in melanoma and UC [10,15], which further provided supports for our observations. Among ccRCC patients, we found the scores of CIBERSORT.CD8 and gene.CD8 in “Tumor-infiltrating lymphocyte (TIL)” category showed significant correlations with clinical response, achieving the best performance superior to the other biomarkers. These findings supported by previous clinical and genomic studies, which demonstrated that ccRCC was a highly immune-infiltrated tumor and featured the increased immune signature [16]. While in NSCLC patients, we found that only the immune checkpoints PD-L2 and CTLA-4 suggested preferable performance than other biomarkers. In general, our results revealed that most of biomarkers showed different performance to ICB response in various cancer types, while some had the preference in specific cancer types, suggesting different cancer underlined markedly distinct immuno-oncology interaction mechanisms with ICB blockade. Nonetheless, the immune checkpoint biomarkers shared among these four cancer types, and the “Effector molecule” biomarkers especially IFN-gamma also presented accordantly predictive ability to ICB response, implying their well universality and applicability across the cancer types.

Furthermore, we evaluated the performance of response biomarkers under different ICB therapy strategies. Due to the limited availability of sufficient datasets from other cancer types to cover multiple treatment strategies, we only split melanoma patients according to treatment strategies. For anti-PD-1 immunotherapy, immune checkpoint PD-L1, gene.CD8, the biomarkers Pan−F−TBRS, C-ECM-up in “Immune resistance” category and TIDE presented significant association with clinical response and prognostic efficiency. While under the anti-CTLA-4 therapy, our analysis revealed CTLA-4 and IMPRES, TIS, CRMA, and IPS performed better than other biomarkers for predicting treatment response. Interestingly, the top-performing transcriptomic biomarkers under PD-1 blockade couldn’t perform well as those in anti-CTLA-4 therapy and vice versa. These findings were supported by previous studies showing that the CTLA-4 pathway and PD-1 pathway executed non-redundant co-inhibitory roles [51]. For “anti-CTLA-4 prog anti-PD-1” therapy, the biomarkers PD-L2, Expanded immune signature and TIDE showed consistently better performance than the others. Comparing with monotherapies and sequential therapy, more transcriptomic biomarkers were found to be correlated with clinical response, predictive power and prognostic efficacy in patients following combined anti-CTLA-4 and anti-PD-1 therapy [65,66]. Targeting both of the immunosuppressive pathways simultaneously, combination therapy could more effectively, at least in part, induce the infiltration of immune cells, production of inflammatory cytokines and enhancement of tumor antigen presentation [52]. Correspondingly, these notions were consistent with our observations that these immunobiologic process-related biomarkers TIS, IFN−gamma and APM all exhibited significant association with clinical response and achieved better accuracy in combination therapy cohorts.

Due to the association of TME components with response to ICB was frequently reported [53], we invesgated the the associations with clinical ICB response and predictive performance of various TME componets estimated by MCP-counter [29]. In general, the abundance of CD8 T cells and T cells showed significant correlations with clinical response and good predictive efficiency across melanoma, ccRCC and UC. And for patients treated with combined anti-CTLA-4 and anti-PD-1 therapy, most immune cells obtained relatively better prediction performance. In addition, we also found the abundance of the B cell demonstrated positive correlation with ICB response for UC patients, and also achieved preferable predictive performance in ccRCC and in melanoma patients treated with anti-CTLA-4 therapy. These results were consistent with recent studies that B cells were associated with immunotherapy response to ICB in patients with metastatic melanoma, RCC and soft-tissue sarcoma [54,55].

Taken together, these results surprised us in that a few biomarkers showed a tendency of stable association with ICB response in majority of the datasets and the biomarkers showed substantial variation in prediction performance across datasets. Similar to our results, Auslander et al. [13] compared the predictive accuracy of IMPRES with that of current transcriptome-based biomarkers and found the prediction performance fluctuated widely across datasets except IMPRES. While IMPRES was then argued about its statistical validity and generalizability [67,68]. Jiang et al. [11] also compared TIDE with other biomarkers, and showed the biomarker IPS did not achieved the claimed accuracy when using the source codes. In our opinion, there were some other potential factors that could influence the performance of biomarkers even in the specific cancer type and ICB therapy. Currently, we didn’t investigate suitable thresholds of biomarkers for better stratification of patients. While previous evidences showed that more stringent threshold might improve the predictive power of some given biomarkers. For example, IMPRES manifested the best tradeoff between precision and recall under the threshold 8 [13], patients with higher levels of TMB showed better prognostic efficacy [10,69]. These observations suggested that different thresholds might affect the prediction performance of biomarkers and more precise threshold need to be further studied. Another possible factor was the intra- and inter-tumor heterogeneity of tumor microenvironment. Whether a tumor was “hot” or “cold” and the TMB differences really make sense, and these hypotheses need to be further validated. As individual biomarker seemed not be robust for patients stratification, we supposed the combination of multiple biomarkers might perform better than individual biomarkers. There already were some explorations. For example, the combination of CD8+ effector T cell signature and TMB provided significantly improved predictive power over either biomarkers in isolation [10]. We expect further investigation about integrated biomarkers would facilitate more accurate and robust prediction of ICB response in the future.

The potential limitation of our study was the lack of large-scale patient cohort for more sufficient assessment. In our study, we compiled numerous public ICB treatment datasets with variable sample size (each more than 15 patients) and focused on multi-layered analysis. Importantly, the study might suffer from a lack of statistical power due to small samples after stratification of some cohorts. To address this issue, we used the prediction score to take into account sample size differences for the predictive performance evaluation. Besides, we also noticed that the variation of response rates in our benchmark datasets from different cancer types or even the same cancer type. This was consistent with the recent findings that objective response rates varied among different cancer types and ICB therapy strategies, respectively [45,56]. Moreover, current benchmark datasets were from different transcriptomic technologies, both RNA-seq and Nanostring nCounter. As Nanostring nCounter only detects hundreds of pre-selected genes, we carefully checked the genes detected by the Nanostring nCounter and the genes involved in each biomarker. For those biomarkers that not all genes were measured, they would not be evaluated. Therefore, the biomarkers calculated based on expression of a single gene or integration of geneset expression (such as CYT, IFN-gamma) would not be affected under different transcriptomic technologies. Although we tried to address these issues due to divergent public evaluation datasets, we expected ongoing and future more labeled patient ICB response data and large-size prospective clinical trials, it is definitely necessary for further determining an unbiased benchmark and performing an objective biomarker evaluation.

## 5. Conclusions

In summary, based on the systematic benchmark, we assessed the associations with clinical ICB response, predictive performance and prognostic values of current transcriptomic biomarkers at the overall evaluation situation or considering different situations under different cancer types and ICB therapies. Our analysis demonstrated that the different predictive biomarkers exhibited different performance for ICB response across various cancer types and ICB therapy strategies, whereas some performed better than other biomarkers only in specific situation. Nevertheless, there remained to be some benchmark datasets did not capture any significant correlations between predictive biomarkers and ICB response, suggesting limitations of individual biomarkers require the combinations of multiple biomarkers for ICB response prediction in the future. Our study provided a guidance for ICB response biomarker selection and laid the foundation for precision immuno-oncology field.

## Figures and Tables

**Figure 1 cancers-13-01639-f001:**
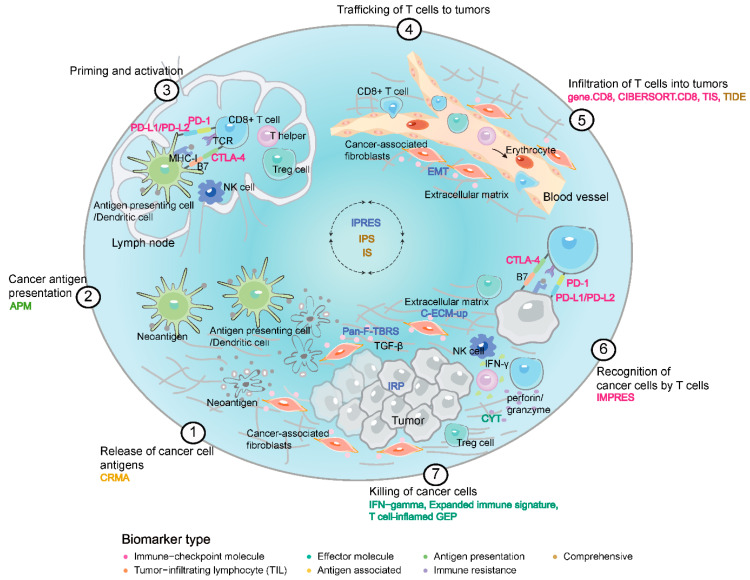
Transcriptomic biomarkers of ICB response in Cancer-immune Cycle.

**Figure 2 cancers-13-01639-f002:**
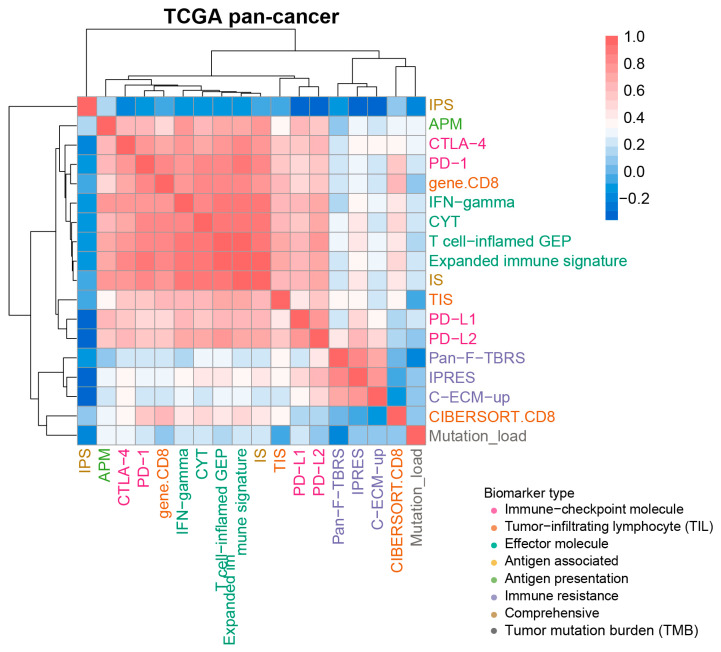
Correlations of transcriptomic biomarkers with ICB response at overall evaluation level. A heatmap displayed the Spearman rank correlation coefficient between any two biomarkers, and demonstrated the hierarchical clustering pattern of various biomarkers based on ~6600 samples from the TCGA pan-cancer cohort. Positive and negative correlations were represented in red and blue, respectively. Biomarkers such as EMT, CRMA, IRP, TIDE and IMPRES were excluded due to them only apply to specific cancer types.

**Figure 3 cancers-13-01639-f003:**
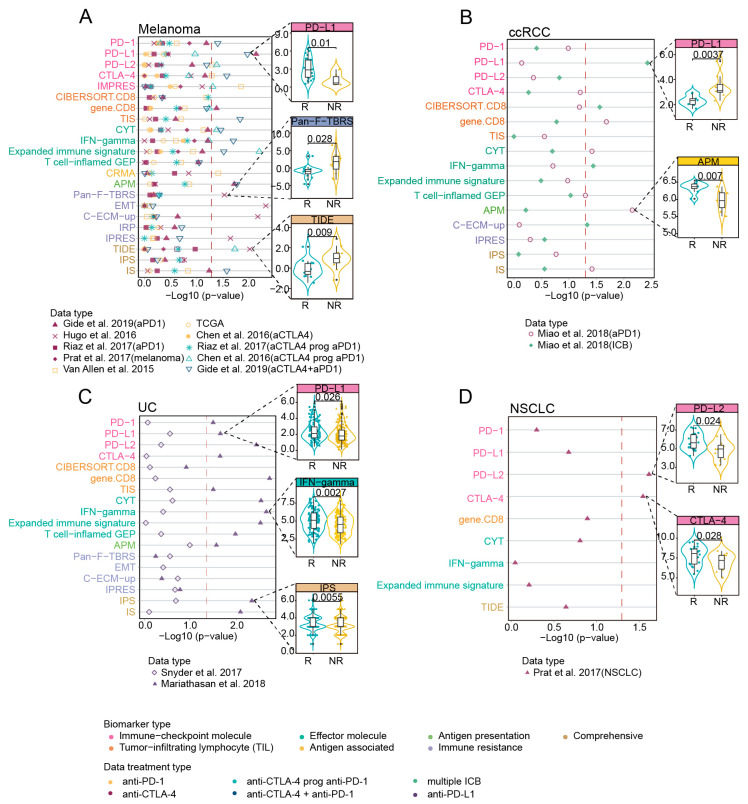
Correlation of biomarkers with clinical response to ICB across multiple datasets with different cancer types. (**A**–**D**) Left: the two-sided Wilcoxon rank-sum test p value indicating whether biomarkers significantly differentiate between responders versus non-responders (NR) (patients stratification using “PD” strategy) in distinct tumors, red dashed line indicated 0.05 threshold of *p* value. The color of the dot represented ICB treatment types of each dataset. Magenta denoted anti-PD-1, yellow denoted anti-CTLA-4, light green denoted “anti-PD-1 after progression on prior anti-CTLA-4”, blue represented combined anti-PD-1 and anti-CTLA-4 immunotherapy, grass green represented multiple ICB treatments, and purple represented anti-PD-L1. Right: boxplots showing examples of biomarkers with significant difference (Wilcoxon rank-sum test *p* < 0.05) between the responding (R) versus non-responding (NR) tumors. Black lines in the box represented upper 75%, median, and lower 25% values.

**Figure 4 cancers-13-01639-f004:**
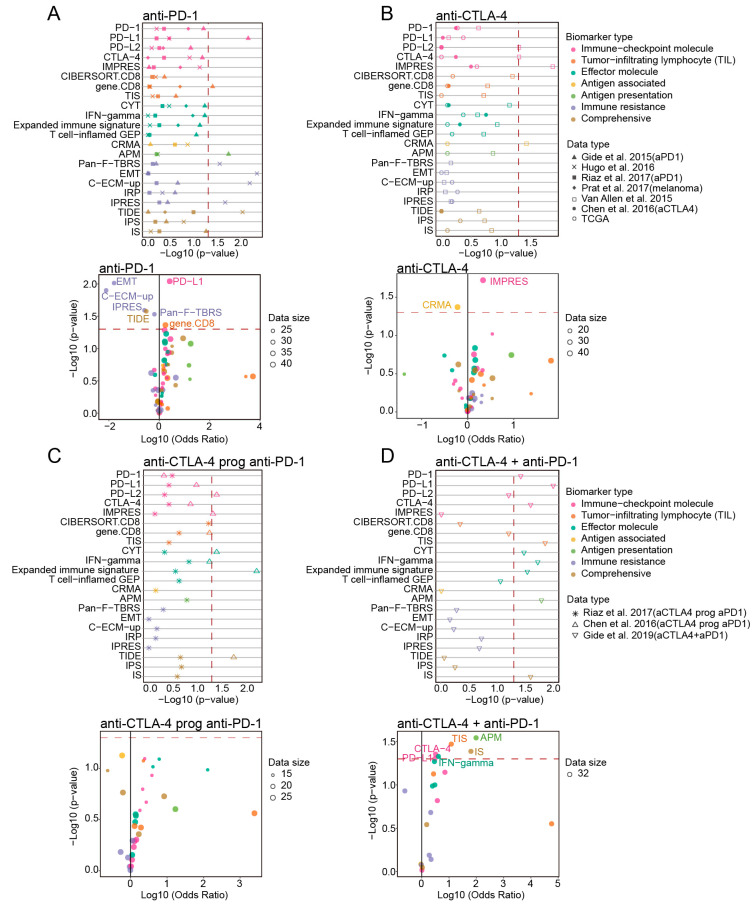
Correlation of biomarkers with clinical response to ICB across multiple datasets with different ICB therapy strategies. (**A**–**D**) Top: the two-sided Wilcoxon rank-sum test *p* value indicating whether biomarkers significantly differentiate between responders (R) versus non-responders (NR) (patients stratification using “PD” strategy) in four ICB therapy strategies, red dashed line indicated 0.05 threshold of *p* value. Bottom: scatterplot showed log10 (Odds Ratio) and −log10 (*p*-value) from univariate logistic regression model for different ICB therapies. Red dashed line indicated 0.05 threshold of *p* value.

**Figure 5 cancers-13-01639-f005:**
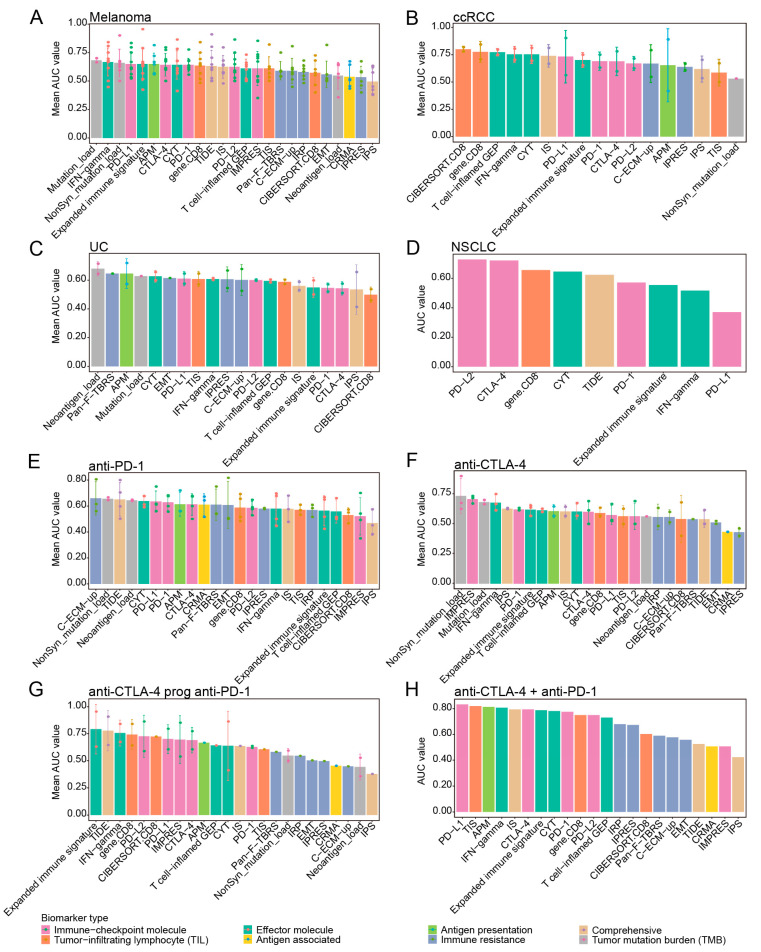
Prediction performance of biomarkers for ICB response across different cancer types and different ICB therapy strategies. (**A**–**D**) Bar centre was defined by the mean AUC values of each transcriptomic biomarker across different cancer datasets (patients stratification using “PD” strategy), and error bars indicated ±1 SD. (**E**–**H**) Bar centre was defined by the mean AUC values of each transcriptomic biomarker in melanoma datasets under different ICB therapy strategies (patients stratification using “PD” strategy), and error bars indicated ±1 SD.

**Figure 6 cancers-13-01639-f006:**
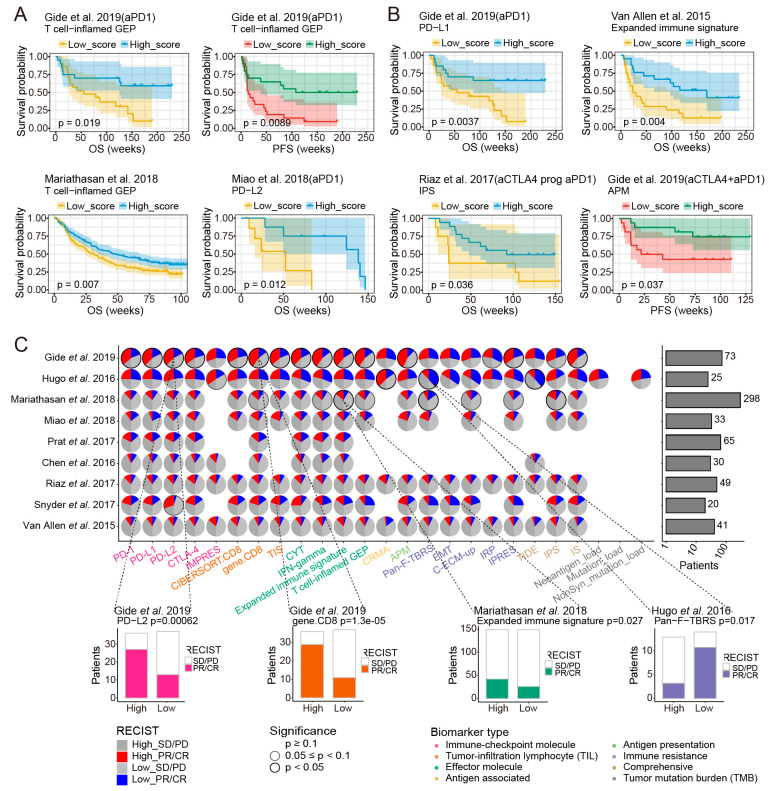
The impact of ICB response biomarkers on clinical efficacy of ICB therapy. Kaplan–Meier survival curves showing OS or PFS for patients with high versus low scores of (**A**) T cell-inflamed GEP in melanoma, T cell−inflamed GEP in UC and PD−L2 in ccRCC. Kaplan-Meier survival curves showing OS for patients with high versus low scores of (**B**) PD−L1 under anti-PD-1 therapy, Expanded immune signature under anti-CTLA-4 therapy and IPS under “anti-CTLA-4 prog anti-PD-1” therapy. Kaplan–Meier survival curves showing PFS for patients with high versus low scores of APM under the combination anti-PD-1 and anti-CTLA-4 therapy. (**C**) Top: the pies on the left panel showing the significance of association for each response biomarker in each overall benchmark dataset and the barplots on the right panel showing the numbers of patients in each dataset. The left half of the pie chart represented the patients with high scores of the corresponding biomarkers, red and dark gray indicated the proportion of responders and non-responders in high-score patients, respectively. The right half of the pie chart represented the patients with low scores of the corresponding biomarkers, blue and light gray indicated the proportion of responders and non-responders in patients with low-score patients, respectively. Borders with no color, gray borders and black borders represented *p* ≥ 0.1, 0.05 ≤ *p* < 0.1, *p* < 0.05, respectively. Different categories of biomarkers were represented by different colors. Bottom: objective response in patients with high versus low scores of PD-L2 and gene.CD8 in the Gide et al., 2019 dataset, Expanded immune signature in the Mariathasan et al., 2018 dataset and Pan−F−TBRS in the Hugo et al., 2016 dataset. Proportion of PR/CR were colored on histograms based on the categories of corresponding biomarkers, with the numbers of patients shown in each bar. Datasets with less than 20 samples were excluded from the analysis.

**Figure 7 cancers-13-01639-f007:**
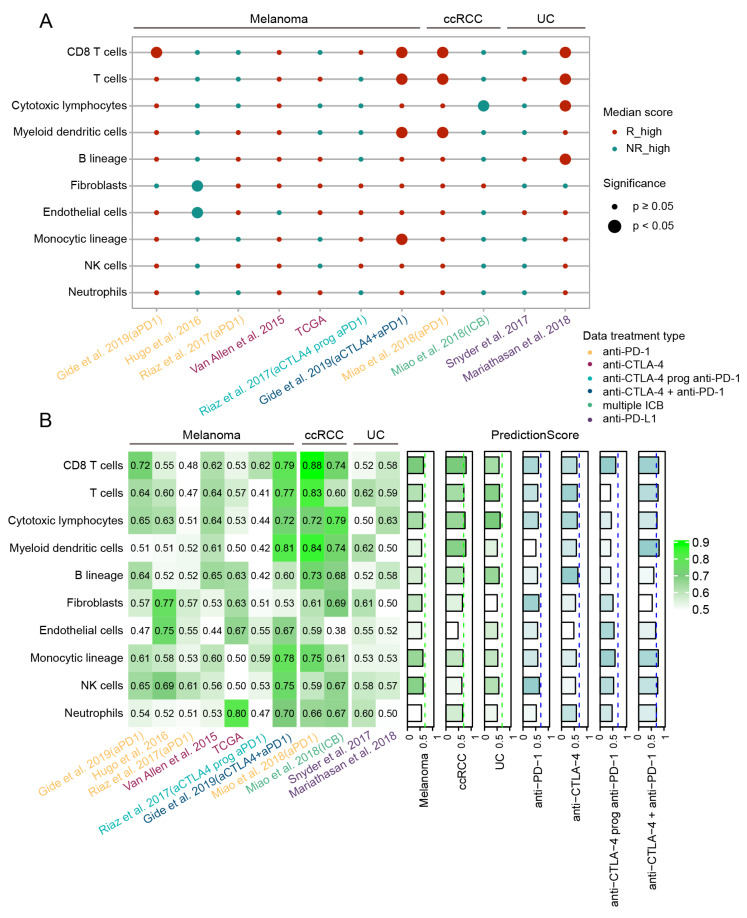
Evaluation of the association between the TME components and clinical response to ICB in different cancer types and therapies. (**A**) Scatterplot showed correlation of the abundance of TME components with ICB response in specific cancer type with different ICB therapy. Red dots denoted that the abundance of TME component was higher in responders than non-responders, green dots indicated the opposite. Size of dots indicated significance (larger for *p* < 0.05) and *p* value was computed by the two-sided Wilcoxon rank-sum test. Colors of different datasets indicated different ICB treatment strategies. (**B**) Prediction performance of each TME component for ICB response across different cancer and treatment types. AUC for each component across benchmark datasets were shown at left panel. The cancer type-specific and treatment type-specific prediction scores (sum of sample size-weighted AUC) were shown at right panel. The gradient of color indicated the prediction performance from low (light) to high (dark).

**Table 1 cancers-13-01639-t001:** Benchmark datasets for the evaluation in different cancer types and ICB therapy strategies.

Data Name ^1^	Data Name ^2^	Tumor Type	Sample ^#^	Objective Response Rate *	Therapeutic Agent	Data Type	RECIST	Reference
Gide et al., 2019	Gide et al., 2019 (aPD1)	Melanoma	41	46.3%	anti-PD-1 (pembrolizumab/nivolumab)	RNA-seq	RECIST 1.1	[20]
	Gide et al. 2019 (aCTLA4 + aPD1)		32	65.6%	anti-PD-1 + anti-CTLA-4 (ipilimumab)			
Riaz et al., 2017	Riaz et al., 2017 (aPD1)	Melanoma	25	26.1%	anti-PD-1 (nivolumab)	RNA-seq	RECIST 1.1	[19]
	Riaz et al., 2017 (aCTLA4 prog aPD1)		26	15.4%	anti-CTLA-4 progression + anti-PD-1 (nivolumab, ipilimumab)			
Van Allen et al., 2015	Van Allen et al., 2015	Melanoma	42	17.1%	anti-CTLA-4 (ipilimumab)	RNA-seq	RECIST 1.1	[22]
Chen et al., 2016	Chen et al., 2016 (aCTLA4)	Melanoma	16	26.7%	anti-CTLA-4 (ipilimumab)	NanoString nCounter	RECIST	[23]
	Chen et al., 2016 (aCTLA4 prog aPD1)		16	6.7%	anti-CTLA-4 progression + anti-PD-1			
Hugo et al., 2016	Hugo et al., 2016	Melanoma	27	55.6%	anti-PD-1 (pembrolizumab/nivolumab)	RNA-seq	irRECIST	[18]
TCGA	TCGA	Melanoma	18	36.4%	anti-CTLA-4 (ipilimumab)	RNA-seq	RECIST	
Prat et al., 2017	Prat et al., 2017 (melanoma)	Melanoma	25	36%	anti-PD-1 (pembrolizumab/nivolumab)	NanoString nCounter	RECIST 1.1	[24]
	Prat et al., 2017 (NSCLC)	NSCLC	35	25.37%				
		HNSCC	5					
Mariathasan et al., 2018	Mariathasan et al., 2018	UC	298	22.8%	anti-PD-L1 (atezolizumab)	RNA-seq	RECIST	[10]
Snyder et al., 2017	Snyder et al., 2017	UC	26	35%	anti-PD-L1 (atezolizumab)	RNA-seq	RECIST 1.1	[25]
Miao et al., 2018	Miao et al., 2018 (aPD1)	ccRCC	16	18.8%	anti-PD-1 (nivolumab)	RNA-seq	RECIST 1.1	[26]
	Miao et al., 2018 (ICB)		17	29.4%	anti-PD-L1 (atezolizumab)/anti-PD-1 + anti-CTLA-4 (nivolumab, ipilimumab)			

^1^ Ten benchmark datasets for overall analysis. ^2^ Fifteen datasets for cancer-specific and treatment-specific analysis. ^#^ Total number of patients included in the dataset. Among them, there are a few of patients missed response information that were excluded in analysis. * Objective response rate (ORR) = (CR + PR)/total patients.

**Table 2 cancers-13-01639-t002:** Summary of 22 transcriptomic predictive biomarkers included in this study for ICB response prediction.

Biomarker Category	Biomarker	Definition	Correlation to ICB Response	Data Type	Tumor Type	Reference
Immune-checkpoint molecule	PD-1	PD-1 mRNA expression	Positive	RNA-seq/NanoString nCounter/MicroArray	Multiple Cancer Types	[35,36]
Immune-checkpoint molecule	PD-L1	PD-L1 mRNA expression	Positive	RNA-seq/NanoString nCounter/MicroArray	Multiple Cancer Types	[8,37]
Immune-checkpoint molecule	PD-L2	PD-L2 mRNA expression	Positive	RNA-seq/NanoString nCounter/MicroArray	Multiple Cancer Types	[36,38]
Immune-checkpoint molecule	CTLA-4	CTLA-4 mRNA expression	Positive	RNA-seq/NanoString nCounter/MicroArray	Multiple Cancer Types	[8]
Immune-checkpoint molecule	IMPRES (immuno-predictive score)	A predictor of ICB response in melanoma which encompasses 15 pairwise transcriptomics relations between immune checkpoint genes.	Positive	RNA-seq/NanoString nCounter/MicroArray	Melanoma	[13]
Tumor-infiltrating lymphocyte (TIL)	CIBERSORT.CD8	Fraction of tumor- infiltrating CD8 + T cells	Positive	RNA-seq/MicroArray	Multiple Cancer Types	[17]
Tumor-infiltrating lymphocyte (TIL)	gene.CD8	The mean expression of CD8A and CD8B genes/the CD8 gene expression	Positive	RNA-seq/NanoString nCounter/MicroArray	Multiple Cancer Types	[17]
Tumor-infiltrating lymphocyte (TIL)	TIS (T Cell Infiltration Score)	T cell infiltration score about nine T cell subsets	Positive	RNA-seq/MicroArray	Multiple Cancer Types	[16]
Effector molecule	CYT (immune cytolytic activity)	Immune cytolytic activity	Positive	RNA-seq/NanoString nCounter/MicroArray	Multiple Cancer Types	[39]
Effector molecule	IFN-gamma	IFN-γ 10-gene expression	Positive	RNA-seq/NanoString nCounter/MicroArray	Multiple Cancer Types	[15]
Effector molecule	Expanded immune signature	IFN-γ 28-gene expression	Positive	RNA-seq/NanoString nCounter/MicroArray	Multiple Cancer Types	[15]
Effector molecule	T cell-inflamed GEP(gene expression profiles)	IFN-γ–responsive genes expression	Positive	RNA-seq/NanoString nCounter/MicroArray	Multiple Cancer Types	[15]
Antigen associated	CRMA (anti-CTLA-4 resistance associated MAGE-A)	MAGE-A genes expression	Negative	RNA-seq/NanoString nCounter/MicroArray	Melanoma	[40]
Antigen presentation	APM (Antigen Presenting Machinery)	Seven antigen presenting machinery (APM) genes expression	Positive	RNA-seq/MicroArray	Multiple Cancer Types	[16]
Immune resistance	Pan_F_TBRS (pan-fibroblast TGF-β response signature)	Pan-fibroblast TGF-β response signature	Negative	RNA-seq/NanoString nCounter/MicroArray	Multiple Cancer Types	[10]
Immune resistance	EMT (epithelial–mesenchymal transition)	Epithelial–mesenchymal transition (EMT)-related gene expression	Negative	RNA-seq/NanoString nCounter/MicroArray	Melanoma, UC	[18,41]
Immune resistance	C-ECM-up (cancer-associated extracellular matrix genes upregulated)	Cancer-associated ECM upregulated genes enrichment score	Negative	RNA-seq/MicroArray	Multiple Cancer Types	[30]
Immune resistance	IRP (ImmuneResistanceProgram)	Resistance program that is associated with T cell exclusion and immune evasion	Negative	RNA-seq/MicroArray	Melanoma	[42]
Immune resistance	IPRES (Innate Anti-PD-1 RESistance)	Innate anti-PD-1 resistance gene signature	Negative	RNA-seq/MicroArray	Multiple Cancer Types	[18]
Comprehensive	TIDE (Tumor Immune Dysfunction And Exclusion)	A computational method to model two primary mechanisms of tumor immune evasion	Negative	RNA-seq/NanoString nCounter/MicroArray	Melanoma, NSCLC	[11]
Comprehensive	IPS (Immunophenoscore)	A scoring scheme for the quantification termed immunophenoscore.	Positive	RNA-seq/NanoString nCounter/MicroArray	Multiple Cancer Types	[12]
Comprehensive	IS (Immune Signature)	Bayesian probability of the immune signature	Positive	RNA-seq/NanoString nCounter/MicroArray	Multiple Cancer Types	[43]

## Data Availability

The datasets used in this study have been published in previous studies, and the detailed data sources can be found in Table 1.

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
