# Peer review of "Systematic Assessment of Transcriptomic Biomarkers for Immune Checkpoint Blockade Response in Cancer Immunotherapy"

_cancers, 2021, doi:10.3390/cancers13071639_

Round 1
Reviewer 1 Report
This manuscript offers novel insights into the issue of identifying biomarkers suitable for detecting patients that benefit from ICB treatment. The design of the study is based on bioinformatical and biostatistical analysis of expression data from multiple datasets. In my humble opinion the biostatistical analysis are all sound and presented in nice figures and graphs that clearly show the results to a less specialized audience.
That said, I do have some remarks concerning this manuscript.
Major remarks:
The analysis is based on multiple datasets. However, the statistical analysis is not clear cut. Whereas some datasets show a significant p.value for a specific marker (e.g. CYT), other datasets do not show a significant value. There was no clear explanation in the manuscript explaining this phenomenon. If a biomarker is (clinical) relevant, one would expect it to be significant in the majority of the datasets. This issue is present in the overall analysis , in the analysis of different cancer types and in the analysis of different ICB therapy strategies.
The design of the manuscript relies completely on in silico analysis and lacks confirmation/validation of the result (s.a. (in vitro) with another technique or on a validation set (training set à validation set)). In that sense, the analysis is not finished.
At present, the manuscript is quite long and lacks a direction / main message.
Minor remarks:
- Introduction, pag 2. “The results showed that TIE achieved consistently better performance for both anti-PD-1 and anti-CTLA-4 therapies.” Better performance than ????
- Introduction, pag2. IPS abbreviation is mentioned for first time without explanation
- Material and methos:16 caner types are all mentioned by abbreviation. Please also provide full length names.
- Table 2: Prat et al, 2017 dataset consists of Mel, NSCLC and HNSCC. Please supply the amounts of each tumor type separately.
- Table 2: would it not be more convenient if you order the 10 studies by tumor type. That way the reader can see how many melanoma, UC, RCC and NSCLC patients are included.
- “3.5 Evaluating the association …” Please mention the amount of cancer patients which were evaluated. The RCC and NSCLC datasets are very small, this should be addressed.
- The overall analysis where all the data of the different tumor types and different treatment strategies are mixed together is not so interesting. The data in which the cancer types are split is much more relevant. The split in treatment strategies is much more artificial. A further split of the therapies in the different cancer types would even be more interesting (as for example CTLA-4 treatment is not clinical relevant in all examined cancer types).
- As none of the 22 markers on its one is significant relevant in the majority of datasets, would a combination of markers (e.g. 2-3 markers from the 7 different marker categories) not be more suited/powerful as candidate biomarker. Has this been explored?
- A table with a clear overview of the relevant markers in the different settings (overall, cancer type specific, treatment specific) would benefit the manuscript. This way, the main observations and main message of this study might become more visible to the audience.
Reviewer 2 Report
Sun et al. present a benchmark to assess to relevance of several previously published biomarkers for response to immune checkpoint inhibitors in various cancer types and therapeutic options. This is an important topic since often biomarkers are hard to replicate. However, the current benchmark is far from being exhaustive as the authors claim and is not presented in a very clear way. Here are some points that would need to be addressed before this manuscript is suitable for publication:
Major points:
1) The authors repeatedly claim their benchmark is comprehensive. This is not the case and it cannot be the case. They only include some cancer types, for which public datasets are available, and only include some biomarkers, not all possible. This claim needs to be modified.
2) Some important biomarkers are not present in this benchmark work and should be added. Notably, in the tumor microenvironment (TME), only CD8+ T cells are included via the CIBERSORT estimate. However, most TME components have been shown to be associated with response or resistance to immune checkpoint blockade (see DOI 10.3389/fimmu.2020.00784). In particular, B cells have been shown to favor response to ICB in melanoma, ccRCC and soft-tissue sarcoma (3 back-to-back articles in Nature, January 2021). The evaluation of ME populations as biomarkers of response needs to be more thorough.
3) The use of CIBERSORT to evaluate the TME composition between various samples is inaccurate. Indeed, it returns abundances of the populations within the overall immune infiltrate, not within the whole sample. Other deconvolution methods are much more fitted to this purpose, such as MCP-counter or EPIC.
4) It is unlikely to have universal biomarkers that would perform well in all cohorts for all cancer types and all immunotherapies options. Therefore, the authors should put more focus on what markers are cancer-type or therapy-dependent. Moreover, types of treatment should be indicated by some sort of color or shape code in figure 4, and cancer type should be indicated similarly on figure 5.
5) In the current state, the manuscript is unclear and the conclusions drawn by the authors are hard to follow. There are several points that, to my opinion, blur the message and could be improved:
5a) Some supplementary figures (for instance, S2, S7, S8, S21, S26 and S27) have far too much information on them to be humanly readable. The authors should find other data visualization tools to make their data easier to comprehend. Maybe they could consider using heatmaps for numerical data, forestplots for survival data, … For instance, Fig 7 A to I could be replaced by a single forestplot, easier to read. Moreover, how to read and interpret Figure 7J is unclear to me.
5b) Figures 2 and 3 mix too many things (different cancer types, therapeutic options, …) and they are therefore too complicated. However, the same data is displayed in a more precise manner in figure 4, 5 and 6. Therefore, figures 2 and 3 appear to be redundant and the authors could consider to remove them altogether. This would make their point more concise and readable.
5c) The author could consider adding a summary table that would give, for all biomarkers considered, a “rank” for each cancer type and treatment option. To do so would require to assign scores to the methods based on the different metrics used by the authors, but would be a nice addition to the manuscript. This would give readers a clear picture of the findings in a glance.
6) The datasets have extremely variable sample size. This needs to be discussed (statistical power) and included in the metrics: the authors need to ponderate some metrics by the sample size, in order to give more weight to the larger datasets and not to overinterpret small datasets.
7) The data included here come from diverse technologies. This needs to be discussed. Indeed, Nanostring nCounter only allows a few hundreds of genes to be measured while RNA-seq potentially analyses the full transcriptome, and this has strong repercussion onto the possibility to discover biomarkers.
8) How was the literature search conducted to identify the 22 biomarkers? To show it is somewhat comprehensive, the authors should provide the details of the query and on what literature database is has been conducted. The inclusion or exclusion criteria must also be provided. Otherwise, this cannot be called a comprehensive search, and the study is simply a benchmark of the most common biomarkers. This is fine, but is should be labeled this way and not oversold.
Minor points:
1) “Nanostring” is a company, not a transcriptomics technology. The technology is called “Nanostring nCounter”.
2) The response rates observed in the datasets should be included in table 2, for the various ways to measure response. The variabilities of response rates should also be discussed.
3) Tables 2 and 3 are very redundant. Perhaps they could be combined into one table to make the manuscript more concise.
Reviewer 3 Report
The manuscript provide in-depth and insightful comprehensive data from public regarding the IO treated multiple cancer subtypes.
the authors presented many data, and do really comprehensive analysis.
however, I think there are several major issues:
- how the author selected the "biomarkers" to study?
- so many have been selected (randomly)? and multiple testing correction will be a mess.
- some of the figures are totally not readable, all the lines and dots are squeezed together, and too many colors (some looks almost identical) and i have been wondering which one is significant and which one is not throughout the entire manuscript.
- i suggest the author has a summary table to show each biomarkers significant in what kinda analysis, which cancer and what study.
- find some of the most significant (especially most universal) and prove it and discuss extensively in discussion.
Reviewer 4 Report
In this study, the authors categorized transcriptomic immune markers for responses to immune checkpoint blockade (ICB), identified their relationship, and assessed their predictive performance by the following criteria:
- Association with response groups (odds ratio and significance): Fig. 2B, and 2C.
- Predictive power for clinical response to ICB (AUC of ROC): Fig. 3
- Preference of specific cancer types, evaluated by association with response groups: Fig. 4
- Preference of specific therapies, evaluated by association with response groups: Fig. 5
- Preference of specific cancer types and therapies, evaluated by predictive power for clinical response to ICB (AUC of ROC): Fig. 6
- Predictive power for clinical efficacy (survival analysis): Fig. 7
This is a timely important benchmarking study of the available immune markers for cancer responses to ICB. There are two questions for discussion:
- Can increasing the stringency of threshold improve the predictive power of a given biomarker?
- Can the combination of two distinct categories of biomarkers (e.g. TMB and immune resistance) improve the predictive power?
Round 2
Reviewer 1 Report
/
Reviewer 2 Report
The authors have adequately adressed my concerns. I think the overall quality of the manuscript has largely improved.